# Unbounded entanglement production via a dissipative impurity

Vincenzo Alba[1*]

**1** Institute for Theoretical Physics, Universiteit van Amsterdam, Science Park 904, Postbus 94485, 1098 XH Amsterdam, The Netherlands
* v.alba@uva.nl

October 8, 2021

## Abstract

**We investigate the entanglement dynamics in a free-fermion chain initially prepared in a Fermi sea and subjected to localized losses (dissipative impurity). We derive a formula describing the dynamics of the entanglement entropies in the hydrodynamic limit of long times and large intervals. The result depends only on the absorption coefficient of the effective delta potential describing the impurity in the hydrodynamic limit. Genuine dissipation-induced entanglement is certified by the linear growth of the logarithmic negativity. Finally, in the quantum Zeno regime at strong dissipation the entanglement growth is arrested (Zeno entanglement death).**

## 1 Introduction

Common experience suggests that the interaction between a quantum system and its environment, and the ensuing dissipation, is detrimental for quantum entanglement. In recent years this view was challenged as it was realized that dissipation can be a resource to engineer quantum states [1], for quantum computation [2], or to stabilize exotic states of matter, such as topological order [3]. These results, together with the interest in Noisy-Intermediate-Scale-Quantum (NISQ) devices [4], urge for a thorough understanding of the interplay between entanglement and dissipation in open quantum systems.

A major obstacle is that it is a challenging task to encapsulate the system-environment interaction within a theoretical framework. Within the so-called Markovian approximation, the Lindblad equation provides a powerful framework to address open quantum systems [5]. Interestingly, for some models it is possible to obtain exact solutions of the Lindblad equation [6–16], for instance, in noninteracting systems with *linear* dissipators [6]. Perturbative field-theoretical approaches are also available [17]. A promising direction is to extend the hydrodynamic framework to integrable systems subjected to dissipation [14, 18–21]. This is motivated by the tremendous success of Generalized Hydrodynamics (GHD) for integrable systems [22, 23]. In some simple free-fermion setups it has been shown that it is possible to use a hydrodynamic approach to described the entanglement dynamics [24, 25]. This generalizes a well-known quasiparticle picture for the entanglement spreading in integrable systems [26–33].

Dissipative impurities provide a minimal theoretical laboratory to study the effects of dissipation in quantum many-body systems. They are the focus of rapidly-growing interest, both theoretical [34–46], as well as experimental [47–52], also in interacting fermionic systems [53, 54]. The interplay between entanglement and thermodynamic entropy in the presence of dissipative impurities has not been explored much.

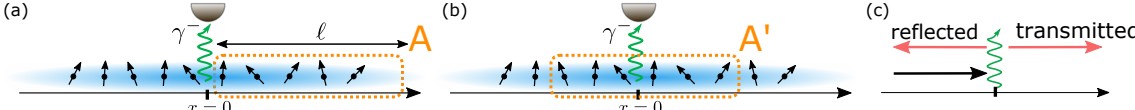

Figure 1: Dissipation-induced entanglement growth. (a) and (b) A free-fermion chain is prepared in Fermi sea and subject to fermionic losses acting at the center of the chain at $x = 0$. Here $\gamma^-$ is the loss rate. We are interested in the entanglement entropy $S$ of a subregion of length $\ell$. We consider two partitions. In the first one (side partition) $A$ is placed next to the dissipative impurity (see (a)). In the second one (centered partition) a subregion $A'$ is centered around the impurity (see (b)). (c) Mechanism for entanglement generation. A fermion reaching the origin can be absorbed or reflected. The reflected and transmitted fermions are entangled.

    One aim of this paper is to start such investigation. We focus on noninteracting fermions with localized fermion losses. The chain is initially prepared in a Fermi sea, and then undergoes Lindblad dynamics. To monitor the entanglement dynamics we consider the entanglement entropies [55–58] (both von Neumann and Rényi entropies), and the fermionic logarithmic negativity [59–75]. The setup is depicted in Fig. 1. An infinite chain is prepared in a Fermi sea with generic Fermi level $k_F$. The dissipation acts at the origin removing fermions incoherently at a rate $\gamma^-$. To quantify the entanglement shared between different subregions we consider the bipartitions of the chain shown in Fig. 1 (a) and (b). In (a) (side bipartition) a subsystem $A$ of length $\ell$ is placed next to the impurity, whereas in (b) (centered partition) a subsystem $A'$ of the same length is centered around the origin. Here we focus on the hydrodynamic limit of large $\ell$ and long times, with their ratio fixed. A crucial observation is that in the hydrodynamic limit of large distances from the dissipation source and long times, the dissipation acts as an effective delta potential (dissipative impurity) with imaginary strength. The associated reflection and transmission amplitudes can be derived analytically [44]. The presence of loss dissipation is reflected in a nonzero absorption coefficient.

    Due to the nonunitary dynamics entanglement and thermodynamic correlations are deeply intertwined. The origin of entanglement is understood as follows. The mechanism is depicted in Fig. 1 (c). The effective delta potential at the origin gives rise to a superposition between the transmitted and the reflected fermion, which form an entangled pair. The propagation of entangled pairs generate entanglement between different spatial regions of the system. More precisely, regions that share entangled pairs get entangled. A similar mechanism is responsible for entanglement production in free-fermion chains with a defect [76–80]. Together with quantum entanglement, thermodynamic correlation is produced during the dynamics. Although the initial state is homogeneous, dissipation gives rise to a nontrivial density profile. This is accompanied by the creation of thermodynamic entropy. Here we show that the entanglement entropies cannot distinguish between these two types of correlations. The reason is that due to the nonunitary dynamics the total system is not in a pure state and the von Neumann entropy and the mutual information are not proper entanglement measures for globally mixed states. This can be understood physically as follows. One can think of the global mixed state as emerging from a larger system comprising the original system and some environment. The density matrix of the original system is obtained by tracing over the degrees of freedom of the *ad hoc* chosen environment (purification). This trace introduces some correlation between the degrees of freedom of the original system. In contrast, the logarithmic negativity is a proper measure also for globally mixed states, and it does not suffer from this problem. The fact that the

von Neumann entropy is not a good entanglement measure is reflected in a generic linear growth with time. This linear growth in open quantum systems has been observed already, for instance, in [81]. One of our main results is that in the hydrodynamic limit the von Neumann entropy of $A$ (see Fig. 1 (a)) is described by

$$S = \frac{\ell}{2} \int_{-k_F}^{k_F} \frac{dk}{2\pi} H_1(1 - |a|^2) \min(|v_k|t/\ell, 1). \tag{1}$$

We provide similar results for $A'$. In (1) we defined $H_1(x) := -x \ln(x) - (1 - x) \ln(1 - x)$, and $v_k$ is the fermion group velocity. Crucially, $|a|^2$ is the absorption coefficient, which is nonzero because of the losses. For lattice systems a maximum velocity $v_{\max}$ exists and (1) predicts a linear growth at short times $v_{\max}t/\ell < 1$, followed by a volume-law scaling at long times. We provide similar results for the Rényi entropies and the moments of fermionic correlation functions. Formula (1) is similar to that describing the entanglement dynamics in a free-fermion chain with a bond defect [76]. The main difference is that in the unitary case the growth of the entropy depends only on the transmission coefficient of the defect. We should stress that although we present results only for the two geometries in Fig. 1, it should be possible to generalize Eq. (1) to to arbitrary bipartitions.

Again, the linear growth in (1) does not reflect genuine entanglement production, which can be diagnosed by the logarithmic negativity. For instance, we show that the logarithmic negativity grows linearly with time for subsystem $A$, whereas it does not increase for $A'$. This supports the mechanism outlined above. For the bipartition in Fig. 1 entanglement is due to the shared pairs formed by the transmitted and the reflected fermions. On the other hand, for the bipartition in Fig. 1 (b) these pairs are never shared between $A'$ and its complement.

The manuscript is organized as follows. In section 2 we introduce the model and the setup. In particular, we review the formula for the fermionic correlators in the hydrodynamic limit, which are the main ingredients to compute the entanglement entropies and the negativity. These formulas where presented elsewhere [39, 44]. Entangled-related quantities are introduced in section 3. In section 4 we present our main results. We first discuss the formula describing arbitrary functions of the moments of the fermionic correlators in the hydrodynamic limit. In section 4.1 we specialize to the moments of the fermionic correlators. In section 4.2 we discuss the hydrodynamic behavior of the entanglement entropies. In section 4.3 we focus on the stationary value of the entanglement entropy, discussing its dependence on the dissipation strength. In section 5 we present numerical benchmarks. We focus on the moments of the fermionic correlators in section 5.1, and on the entanglement entropies in section 5.2. We discuss some future directions in section 6. In Appendix A we report the derivation of the main result of section 4.

## 2 Localized losses in a Fermi sea: Review of known results

Here we consider the infinite free-fermion chain defined by the Hamiltonian

$$H = \sum_{x=-\infty}^{\infty} (c_x^\dagger c_{x+1} + c_{x+1}^\dagger c_x), \tag{2}$$

where $c_x^\dagger, c_x$ are creation and annihilation operators at site $x$. The fermionic operators obey standard canonical anticommutation relations. To diagonalize (2) one defines a Fourier transform with respect to $x$, introducing the fermionic operators $b_k$ in momentum space

as

$$b_k := \sum_{x=-\infty}^{\infty} e^{-ikx} c_x, \quad c_x = \int_{-\pi}^{\pi} \frac{dk}{2\pi} e^{ikx} b_k. \tag{3}$$

Eq. (2) is rewritten in terms of $b_k$ as

$$H = \int_{-\pi}^{\pi} \frac{dk}{2\pi} \varepsilon_k b_k^\dagger b_k, \quad \varepsilon_k := 2\cos(k). \tag{4}$$

Eq. (4) is diagonal, and it conserves the particle number. Let us consider a generic fermion density $n_f = k_F/\pi$, with $k_F$ the Fermi momentum. The ground state of (2) is obtained by filling the single-particle states with quasimomenta $k$ in $k \in [-k_F, k_F]$. The state with $n_f = 1$ ($k_F = \pi$) in which all the quasimomenta are occupied is a product state, and it has trivial correlations. For intermediate filling $0 < k_F < \pi$ the ground state of (2) is critical, with power-law correlations.

From the single-particle dispersion in (4) we define the group velocity $v_k$ of the fermions as

$$v_k := \frac{d\varepsilon_k}{dk} = -2\sin(k). \tag{5}$$

Here we consider the out-of-equilibrium dynamics under the Hamiltonian (2) and localized loss processes at the center of the chain. These are treated in the formalism of quantum master equations [5]. The time-evolved density matrix $\rho_t$ of the system is described by

$$\frac{d\rho_t}{dt} = -i[H, \rho_t] + L^- \rho_t L^{-\dagger} - \frac{1}{2}\{L^{-\dagger}L^-, \rho_t\}. \tag{6}$$

Here, the so-called Lindblad jump operator $L^-$ is defined as $L^- = \sqrt{\gamma^-} c_0$ (see Fig. 1 for a pictorial definition), with $\gamma^-$ the loss rate. Eq. (6) describes incoherent absorption of fermions at the center of the chain.

Entanglement properties of the systems can be extracted from the fermionic two-point correlation functions, i.e., the *covariance matrix*

$$G_{x,y}(t) := \mathrm{Tr}(c_x^\dagger c_y \rho(t)). \tag{7}$$

The dynamics of $G_{x,y}$ is obtained as (we drop the dependence on the coordinates $x, y$ to lighten the notation)

$$G(t) = e^{t\Lambda} G(0) e^{t\Lambda^\dagger}, \tag{8}$$

where $G(0)$ is the matrix containing the initial correlations. The matrix $\Lambda$ is defined as

$$\Lambda = ih - \frac{\Gamma^-}{2}, \tag{9}$$

where $h = \delta_{|x-y|,1}$ is the Hamiltonian contribution while $\Gamma^- = \gamma^- \delta_{x,0}$ encodes the localized dissipative effects. The covariance matrix $G_{x,y}$ is the solution of the linear system of equations

$$\frac{dG_{x,y}}{dt} = i(G_{x+1,y} + G_{x-1,y} - G_{x,y+1} - G_{x,y-1}) - \frac{\gamma^-}{2}(\delta_{x,0} G_{x,y} + \delta_{y,0} G_{x,y}). \tag{10}$$

Here we are interested in the hydrodynamic limit of large distances from the origin and long times, i.e., $x, y, t \to \infty$ with the ratios $x/t, y/t$ fixed. In this limit it can be shown that the dissipation is effectively described by a delta potential. The strength of the potential is imaginary, which is a consequence of nonunitarity. Several properties of the system can be derived by studying the scattering problem of a quantum particle with an imaginary delta

potential [82]. For several initial states, both homogeneous as well as inhomogeneous ones, the dynamics of $G_{x,y}$ can be described solely in terms of the initial fermionic occupations and the reflection and transmission coefficients of the emergent delta potential [44]. Here we are interested in the situation in which the initial state of the dynamics is a Fermi sea with arbitrary Fermi momentum $k_F$.

## 2.1 Hydrodynamic limit of the covariance matrix

In the hydrodynamic limit the solution of (10) with initial condition the Fermi sea is obtained as [44] (see also [39])

$$G_{x,y}(t) = \int_{-k_F}^{k_F} \frac{dk}{2\pi} (e^{ikx} + \chi_x(t)r(k)e^{i|kx|})(e^{-iky} + \chi_y(t)r(k)e^{-i|ky|}). \tag{11}$$

Notice the absolute values in the second terms in the brackets. Moreover, one should observe that the contributions associated with the two coordinates $x, y$ factorize. This factorization is crucial [40] to obtain the exact solution of (10). In (11) $r(k)$ is the momentum-dependent reflection amplitude of the effective delta potential describing the dissipation source at the origin. The analytic expression for $r(k)$ and for the associated transmission amplitude $\tau(k)$ are given as [44]

$$r(k) := -\frac{\gamma^-}{2} \frac{1}{\frac{\gamma^-}{2} + |v_k|}, \quad \tau(k) := \frac{|v_k|}{\frac{\gamma^-}{2} + |v_k|}, \tag{12}$$

where $v_k$ is the fermion group velocity defined in (5). Notice that (12) coincide with the reflection and transmission amplitude for a quantum particle scattering with a delta potential with imaginary strength $-i\gamma^-/2$ after redefining [82] $v_k \sim k$. Crucially, since the dynamics is nonunitary one has that

$$|a|^2 := 1 - |r|^2 - |\tau|^2 = \frac{\gamma^-|v_k|}{(\frac{\gamma^-}{2} + |v_k|)^2} > 0, \tag{13}$$

where we defined the absorption coefficient $|a|^2$, which is the probability that a fermion with quasimomentum $k$ is removed at the origin.

The time dependence of the correlator in (11) is encoded in the function $\chi_x$, which is defined as

$$\chi_x := \Theta(|v_k|t - |x|). \tag{14}$$

At $t = 0$ from (11) one recovers the initial correlation of the Fermi sea as

$$G_{x,y}(0) = \frac{\sin(k_F(x-y))}{\pi(x-y)}. \tag{15}$$

To get an idea of the effect of the dissipation, it is instructive to consider the dynamics of the local fermionic density $n_{x,t}$

$$n_{x,t} = G_{x,x}. \tag{16}$$

This is discussed in Fig. 2. We plot $n_{x,t}$ versus the scaling variable $x/(2t)$, showing results for $\gamma^- = 0.5$ and $\gamma^- = 10$. The results are obtained by using (11). We focus on the effects of the localized losses on the initial Fermi sea with $k_F = \pi/2$. As expected, the Fermi seas gets depleted with time and a nontrivial density profile forms around the origin. For $|x/(2t)| > 1$ the effect of the dissipation is not present and one has the initial density $1/2$. The density profile exhibits a discontinuity at the origin. This reflects the presence of an

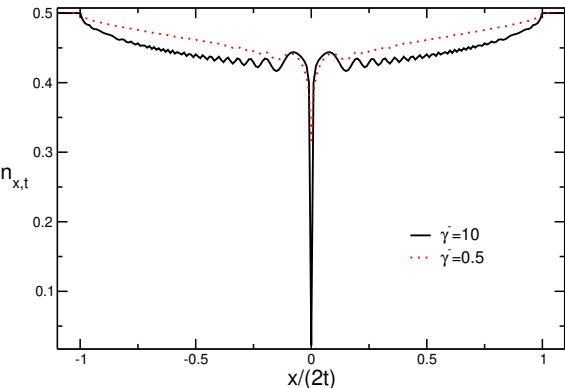

Figure 2: Dynamics of the fermionic density $n_{x,t}$ in the presence of localized losses. Results are for the initial Fermi sea with $k_F = \pi/2$ and for loss rate $\gamma^- = 10$ and $\gamma^- = 0.5$ (continuous and dotted lines, respectively). The oscillations are an artifact of the approximations and vanish in the hydrodynamic limit $x, t \to \infty$ with their ratio fixed. Notice that in the hydrodynamic limit the density develops a discontinuity at the origin.

effective delta potential at the origin. Finally, the oscillations present in Fig. 2 are an artifact of the approximations employed to derive (11), and vanish in the hydrodynamic limit. In the strong dissipation limit $\gamma^- \to \infty$ the evolution of the density freezes. This is a manifestation of the quantum Zeno effect [83–85]. In the following sections we show that the fermionic dynamics shown in Fig. 2 is accompanied with a robust linear entanglement growth with time.

## 3 Entanglement entropies and logarithmic negativity: Definitions

In order to understand how the presence of localized losses affects the entanglement content of the system here we focus on several quantum-information-related quantities, such as the entanglement entropies and the logarithmic negativity. To introduce them, let us consider a bipartition of the system as $A \cup \bar{A}$ (see, for instance, Fig. 1 (a) and (b)). By tracing over the degrees of freedom of $\bar{A}$, which is the complement of $A$, one obtains the reduced density matrix $\rho_A = \mathrm{Tr}_{\bar{A}} \rho$, where $\rho$ is the full-system density matrix. The Rényi entropies are defined as [55–58]

$$S^{(n)} := \frac{1}{1-n} \mathrm{Tr}(\rho_A^n), \quad \text{with } n \in \mathbb{R}. \tag{17}$$

In the limit $n \to 1$ one obtains the von Neumann entropy as

$$S = -\mathrm{Tr}\rho_A \ln(\rho_A). \tag{18}$$

Both Rényi and von Neumann entropies are good entanglement measures provided that the full system is in a pure state. However, in the presence of dissipation the full system is in a mixed state, which introduces some "classical" correlation between $A$ and $\bar{A}$. This spurious, i.e., non-quantum, correlation, affects both the Rényi entropies and the entanglement entropy.

In these situations the so-called logarithmic negativity [60,61] can be used to quantify the amount of genuine entanglement between $A$ and the rest. The logarithmic negativity

$\mathcal{E}$ is defined from the partially-transposed density matrix $\rho^T$. This is defined from $\rho$ by taking the matrix transposition with respect to the degrees of freedom of $\bar{A}$ as

$$\langle e_i, \bar{e}_j | \rho^T | e_k, \bar{e}_l \rangle = \langle e_i, \bar{e}_l | \rho | e_i, \bar{e}_j \rangle, \tag{19}$$

with $e_i, \bar{e}_j$ two bases for $A$ and its complement, respectively. Unlike $\rho$, $\rho^T$ is not positive definite, and its negative eigenvalues quantify the amount of entanglement. The logarithmic negativity is defined as

$$\mathcal{E} = \ln(\mathrm{Tr}|\rho^T|). \tag{20}$$

Here we focus on free-fermion models. For free-fermion and free-boson models both the Rényi entropies and the von Neumann entropy of a region $A$ are calculable from the fermionic correlation function $G_{x,y}$ restricted to $A$, i.e., with $x, y \in A$. Specifically, the Rényi entropies are obtained as [86]

$$S^{(n)} = \frac{1}{1-n} \mathrm{Tr} \ln \left[ G^n + (1-G)^n \right]. \tag{21}$$

In the limit $n \to 1$, one recovers the von Neumann entropy as

$$S = -\mathrm{Tr}(G \ln(G) - (1-G) \ln(1-G)). \tag{22}$$

The logarithmic negativity $\mathcal{E}$ can be calculated efficiently from the two-point function only for free bosons [87]. For free fermions the partial transposed $\rho^T$ is not a gaussian operator, although it can be written as a sum of two gaussian operators [71] as

$$\rho^T = e^{-i\pi/4} O_+ + e^{i\pi/4} O_-, \tag{23}$$

where $O_\pm$ are gaussian operators. Very recently, an alternative definition of negativity has been put forward [73–75]. We dub this alternative negativity *fermionic* negativity. Its definition reads as

$$\mathcal{E} := \ln \mathrm{Tr} \sqrt{O_+ O_-}. \tag{24}$$

Here we use the same symbol $\mathcal{E}$ for the fermionic negativity and for the standard one in (20) because in the following we will only use the fermionic one. In contrast with (20), since the product $O_+ O_-$ is a gaussian operator, the fermionic negativity (24) can be computed effectively in terms of fermionic two-point functions. Specifically, let us rewrite the full-system correlation matrix $G$ as

$$G = \begin{pmatrix} G_{AA} & G_{A\bar{A}} \\ G_{\bar{A}A} & G_{\bar{A}\bar{A}} \end{pmatrix} \tag{25}$$

Here $G_{WZ}$, with $W, Z = A, \bar{A}$ is obtained from the full system $G_{x,y}$ restricting to $x \in W$ and $y \in Z$. One now defines the matrices $G^\pm$ as

$$G^\pm = \begin{pmatrix} -G_{AA} & \pm i G_{A\bar{A}} \\ \pm i G_{\bar{A}A} & G_{\bar{A}\bar{A}} \end{pmatrix} \tag{26}$$

We then define the matrix $C^T$ as

$$C^T = \frac{1}{2} \mathbb{I} - P^{-1}(G^+ + G^-), \quad \text{with } P = \mathbb{I} - G^+ G^-. \tag{27}$$

From the eigenvalues $\xi_i$ of $C^T$ and $\lambda_i$ of $G$ we can define the fermionic negativity $\mathcal{E}$ as [74]

$$\mathcal{E} = \sum_i \left[ \ln[\xi_i^{1/2} + (1-\xi_i)^{1/2}] + \frac{1}{2} \ln[\lambda_i^2 + (1-\lambda_i)^2] \right], \tag{28}$$

It has been shown in Ref. [75] that under reasonable assumptions the fermionic negativity is a good entanglement measure for mixed states.

# 4 Hydrodynamic description of entanglement entropies

We now discuss the out-of-equilibrium dynamics of the entanglement entropies in the hydrodynamic limit. Before that, we provide a more general result, which allows us to obtain the hydrodynamic behavior of the trace of a generic function of the fermionic correlator (cf. (11)). Let us consider the bipartitions in Fig. 1 (a) and (b). In Fig. 1 (a) subsystem $A$ is the interval $[0, \ell]$, i.e., on the right of the dissipative impurity. In Fig. 1 (b) we consider subsystem $A' = [-\ell/2, \ell/2]$ centered around the impurity. Let us consider a generic function $\mathcal{F}(z)$, and let us focus on the quantity $\mathrm{Tr}\mathcal{F}(G_X)$, with $X = A, A'$. In the hydrodynamic limit $t, \ell \to \infty$, with their ratio fixed, one can show that

$$
\mathrm{Tr}\mathcal{F}(G_X) = \ell \int_{-k_F}^{k_F} \frac{dk}{2\pi} \Big[ \Big( 1 - \frac{1}{2z_X} \min(z_X |v_k| t/\ell, 1) \Big) \mathcal{F}(1)
$$
$$
+ \frac{1}{2z_X} \mathcal{F}(1 - z_X |a(k)|^2) \min(z_X |v_k| t/\ell, 1) \Big], \quad z_{A(A')} = 1(2). \quad (29)
$$

Here $v_k$ is the fermion group velocity in (5), and $|a(k)|^2$ is the absorption coefficient of the emergent delta potential (cf. (13)) at the origin. Eq. (29) depends only on $1 - |a|^2$, i.e., the probability of the fermions not to be absorbed at the origin. Also, the only dependence on time is via the factor $\min(z_X |v_k| t/\ell)$, which encodes the fact that $A$ and $A'$ are finite, and information propagates from the origin at a finite velocity $v_k$. The factor $z_X$ in (29) accounts for the different geometries in Fig. 1 (a) and (b), and it has a simple interpretation. For instance, in the argument of the second term in (29) $z_X$ takes into account that for the bipartition in Fig. 1 (a) the number of absorbed fermions is twice that for the bipartition in Fig. 1 (a) because the impurity is at the center of $A'$. Moreover, in $\min(z_X |v_k| t/\ell, 1)$, $z_X$ reflects that for $A'$ the distance between the impurity and the edge of $A'$ is $\ell/2$ instead of $\ell$.

For generic $\mathcal{F}(z)$, Eq. (29) predicts a linear behavior with time for $t \leq \ell/(z_X v_{\max})$, with $v_{\max}$ the maximum velocity in the system. This is followed by an asymptotic saturation at $t \to \infty$ to a volume law $\propto \ell$. Finally, for $\gamma^- = 0$ one recovers the unitary case and from (29), one obtains that

$$
\mathrm{Tr}\mathcal{F}(G_X) = \ell \int_{-kF}^{k_F} \frac{dk}{2\pi} \mathcal{F}(1). \quad (30)
$$

Eq. (30) means that in the absence of dissipation there is no dynamics and for any $\mathcal{F}$ one has a constant contribution that is proportional to $\ell$. The fact that there is no dependence on $z_X$ and on the geometry reflects translation invariance.

The derivation of (29) is reported in Appendix A and it relies on the multidimensional stationary phase approximation [88], and on the assumption that $\mathcal{F}(z)$ admits a Taylor expansion around $z = 0$. We should also stress that although we discuss only the two geometries in Fig. 1 (a) and (b), it should be possible to generalize (29) to arbitrary bipartitions or multipartitions. In the following, by considering different functions $\mathcal{F}(z)$ we provide exact results for the moments of the correlation matrix and the entanglement entropies in the hydrodynamic limit.

## 4.1 Moments of the correlation matrix

Here we study the hydrodynamic limit of the moments $M_n$ of the fermionic correlation matrix. These are defined as

$$
M_n = \mathrm{Tr}(G^n), \quad (31)
$$

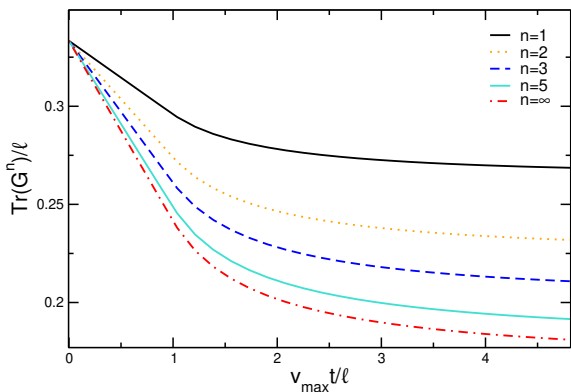

Figure 3: Moments of the fermionic correlator $M_n = \text{Tr}(G^n)$, where $G$ is the fermionic correlation function restricted to subsystem $A$ (see Fig 1). We plot the rescaled moments $M_n/\ell$, with $\ell$ the length of $A$ versus $v_{\max}t/\ell$, $v_{\max}$ being the maximum velocity. Lines are analytic results in the hydrodynamic limit $\ell, t \to \infty$ with their ratio fixed. We only show results for $\gamma^- = 1$ and $k_F = \pi/3$. Note the linear behavior for $t \leq \ell/v_M$ followed by a saturation at $t \to \infty$.

where the correlation matrix $G$ is restricted to subsystem $A, A'$ (see Fig. 1). The behavior of $M_n$ in the hydrodynamic limit is readily obtained from (29) by choosing $\mathcal{F}(z) = z^n$. One obtains that

$$
M_n = \ell \int_{-k_F}^{k_F} \frac{dk}{2\pi} \Big[ \Big( 1 - \frac{1}{2z_X} \min(z_X |v_k| t/\ell, 1) \Big)
$$
$$
+ \frac{1}{2z_X} (1 - z_X |a(k)|^2)^n \min(z_X |v_k| t/\ell, 1) \Big], \quad z_{A(A')} = 1(2). \quad (32)
$$

The structure is the same as in (29). $M_n$ exhibit the same qualitative behavior with a linear decrease at short times $t \leq \ell/v_{\max}$, which is followed by an asymptotic saturation at $t \to \infty$. Several remarks are in order. First, at $t = 0$ one has that for any $n$, $M_n = \ell k_F/\pi$, which is the initial number of fermions in the subsystem. For $t \to \infty$ one has that the number of fermions $M_1$ in the subsystem is

$$
M_1 \xrightarrow{t \to \infty} \ell \int_{-k_F}^{k_F} \frac{dk}{2\pi} \Big( 1 - \frac{|a|^2}{2} \Big). \quad (33)
$$

This means that $M_1 \propto \ell$ for $t \to \infty$, despite the presence of dissipation. In the strong dissipation limit $\gamma^- \to \infty$ one has that $|a|^2 \to 0$, and $M_1 \to \ell k_F/\pi$, i.e., the initial fermion number. This is a manifestation of the quantum Zeno effect. In the limit $\gamma^- \to \infty$ the dynamics of the system is arrested and the number of fermions absorbed at the origin vanishes. Finally, it is interesting to consider $M_n$ in the limit $n \to \infty$. One can readily check that $1 - z|a|^2 < 1$, which implies that only the first term in (32) survives. In particular, in the limit $t \to \infty$, from (32) one obtains that $M_\infty = \ell k_F/\pi(1 - 1/(2z_X))$. For $z_X = 1$ (i.e., for the partition in Fig. 1 (a)) one has $M_\infty = \ell k_F/(2\pi)$, which is half of the initial number of fermions.

In Fig. 3 we show numerical predictions for $M_n$ obtained by using (32). We consider the case with $k_F = \pi/3$ and we restrict ourselves to $\gamma^- = 1$. We provide results only for the bipartition in Fig. 1 (a). The generic behavior outlined above is clearly visible in the figure.

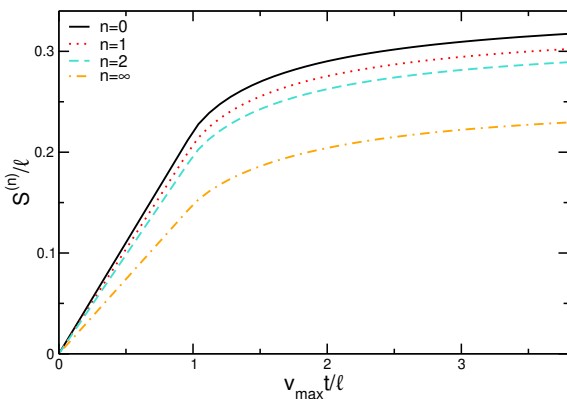

Figure 4: Entanglement entropies $S^{(n)}$ of a subsystem $A$ placed next to the dissipation source (see Fig. 1 (a)). The different lines are analytic predictions in the hydrodynamic limit for different values of $n$. We plot $S^{(n)}/\ell$ versus $v_{\max}t/\ell$, with $\ell$ the size of $A$ and $v_{\max}$ the maximum velocity. We only show results for $\gamma^- = 1$ and $k_F = \pi$.

## 4.2 Entanglement entropies

The hydrodynamic limit of the entanglement entropies, both the von Neumann and the Rényi entropies, is obtained from (29) by choosing

$$\mathcal{F}(z) = H_n(z) = \frac{1}{1-n} \ln[z^n + (1-z)^n]. \tag{34}$$

In the limit $n \to 1$ one recovers the von Neumann entropy by choosing $H_1(z) = -z\ln(z) - (1-z)\ln(1-z)$. After using (34) in (29), and after observing that for any $n$, $\mathcal{F}(1) = 0$, one obtains that

$$S^{(n)} = \frac{1}{2z_X} \frac{\ell}{1-n} \int_{-k_F}^{k_F} \frac{dk}{2\pi} H_n(1 - z_X|a|^2) \min(z_X|v_k|t/\ell, 1). \tag{35}$$

First, for $\gamma^- = 0$, i.e., in the absence of dissipation, one has that $S^{(n)} = 0$ for any $n$. This is consistent with the fact that for a Fermi sea the entanglement entropies exhibit the typical Conformal Field Theory (CFT) logarithmic scaling as [56]

$$S^{(n)} = \frac{c}{6}\left(1 + \frac{1}{n}\right)\ln(\ell) + c_n, \tag{36}$$

where $c = 1$ is the central charge of the model and $c_n$ are nonuniversal constants. The scaling (36) cannot be captured by (35), which describes the leading volume-law behavior $S^{(n)} \propto \ell$. In the strong dissipation limit $\gamma^- \to \infty$, one has that, reflecting the Zeno effect, $S^{(n)}$ vanish for any $n$.

Away from the limits $\gamma^- = 0$ and $\gamma^- \to \infty$, the entanglement entropies increase linearly at short times $t \leq \ell/(z_X v_{\max})$, and saturate to a volume-law scaling $S^{(n)} \propto \ell$ at asymptotically long times. It is interesting to consider the limit $n \to \infty$, which gives the so-called single-copy entanglement. From (35) it is clear that only the first term inside the logarithm in (34) counts, and one obtains that

$$S^{(\infty)} = -\frac{\ell}{2z_X} \int_{-k_F}^{k_F} \frac{dk}{2\pi} \ln(1 - z_X|a|^2) \min(z_X|v_k|t/\ell, 1). \tag{37}$$

It is now crucial to remark that Eq. (35) gives the same qualitative behavior for the entanglement entropies of $A$ and $A'$ (see Fig. 1 (a) and (b)). This is surprising at first because

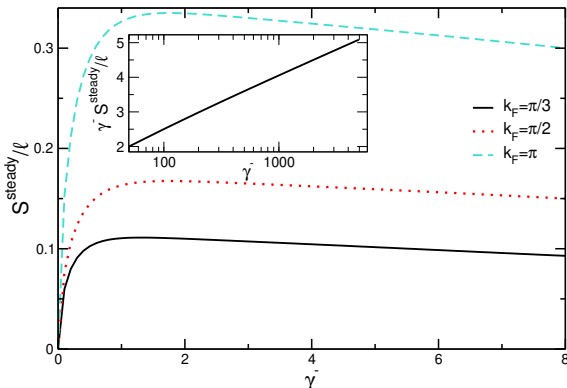

Figure 5: Steady-state entropy in the free fermion chain with localized losses. We show results for the bipartition in Fig. 1 (a). We plot $S^{(\text{steady})}/\ell$ versus the loss rate $\gamma^-$. The different lines in the main figure are for initial states with different Fermi momentum $k_F$. Note that the steady-state entropy has a maximum at $\gamma^- \approx 1$. For $\gamma^- \to \infty$ the steady-state entropy vanishes as $S^{(\text{steady})}/\ell \propto \ln(\gamma^-)/\gamma^-$, as it is shown in the inset.

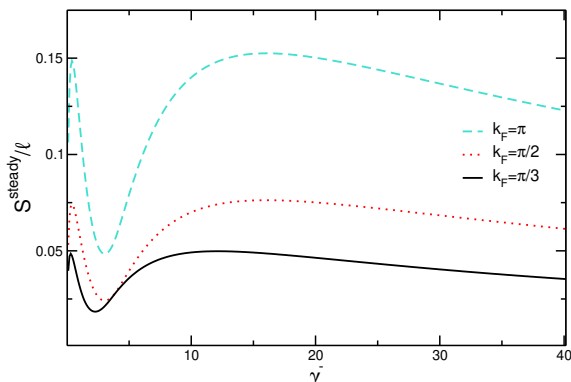

Figure 6: Same as in Fig. 5 for the centered partition in Fig. 1 (b). Notice the presence of two maxima, in contrast with Fig. 5.

no production of entanglement is expected for the centered bipartition in Fig. 1 (b). The reason is that the reflected and the transmitted fermions, which form the entangled pairs, are never shared between $A'$ and its complement. The linear growth in this case should be attributed to the formation of a nontrivial density profile around the origin, which reflects the creation of thermodynamic entropy. The entanglement entropies are not *bona fide* entanglement measures for mixed states because they are sensitive to this thermodynamic contribution. We anticipate that, in contrast, the logarithmic negativity is sensitive to the genuine quantum correlation only (see section 5.2).

In Fig. 4 we report analytic predictions for the dynamics of the entanglement entropies obtained from (35). We plot the rescaled entropies $S_X^{(n)}/\ell$ versus $v_{\text{max}}t/\ell$ for several values of $n$. We consider only the bipartition in Fig. 1 (a), i.e., we choose $X = A$ in (35). Furthermore, we show data for $k_F = \pi$ and $\gamma^- = 1$. The qualitative behaviour discussed above is clearly visible.

### 4.3 Zeno death of entanglement entropy

It is interesting to investigate the steady-state value of the entanglement entropy as a function of the dissipation rate $\gamma^-$. The steady-state entanglement entropy $S^{\text{steady}}$ is obtained from (35) as

$$S^{(\text{steady})} = \frac{\ell}{2z_X} \int_{-k_F}^{k_F} \frac{dk}{2\pi} H_1(1 - z_X|a|^2). \tag{38}$$

In Fig. 5 we plot $S^{(\text{steady})}/\ell$ versus $\gamma^-$. The results are for $X = A$ (see Fig. 1 (a)). In the main plot, the different curves correspond to different values of $k_F$. Notice that the entanglement entropy increases upon increasing $k_F$. This is expected because the entanglement entropy is proportional to the number of fermions that scatter with the impurity at the origin. Interestingly, the data exhibit a maximum in the region $\gamma^- \in [1.5, 2]$. In the strong dissipation limit $\gamma^- \to \infty$ the entanglement entropy vanishes. This is a consequence of the quantum Zeno effect. The decay is as $S^{(\text{steady})} \propto \ln(\gamma^-)/\gamma^-$ (see the inset of Fig. 5).

Finally, it is interesting to compare with the result for the centered partition in Fig. 1 (b). This is discussed in Fig. 6. A richer structure is observed. Indeed, the steady-state entropy exhibits two maxima, one at "weak" dissipation for $\gamma^- \approx 0.5$ and one in the "strong" dissipation regime for $\gamma^- \approx 10$. Notice also that the steady-state entropy is generically smaller than in Fig. 5.

## 5 Numerical benchmarks

We now provide numerical benchmarks for the results derived in section 4. We discuss the moments $M_n$ (cf (32)) in section 5.1. In section 5.2 we focus on the entanglement entropies. Importantly, we discuss the interplay between entanglement and thermodynamic correlation by comparing the evolution of the von Neumann entropy and that of the logarithmic negativity for the two bipartitions in Fig. 1 (a) and (b).

### 5.1 Moments of fermionic correlators

Our numerical results for $M_n$ are discussed in Fig. 7. In the panel (a) and (b) we plot $M_1$ and $M_2$, respectively. We focus on subsystem $A$ (see Fig. 1 (a)). The numerical data in the figure are obtained by using (8). We consider the situation in which the system is initially prepared in a Fermi sea with $k_F = \pi/3$. Notice that $M_1$ is the number of fermions in subsystem $A$. In the absence of dissipation $M_1 = \ell k_F/\pi$ at any time. As a consequence of the fermion loss the number of particle decreases with time. In the figure we report results for several values of $\ell$. Clearly, $M_1$ exhibits the qualitative behavior discussed in Fig. 3. At short times $t \leq \ell/v_{\max}$, $M_1$ decreases linearly, whereas for $t \to \infty$ it saturates. However, the data for finite $\ell$ exhibit sizeable deviations from the hydrodynamic limit result, which is reported as dashed-dotted line in Fig. 7. These deviations are expected. The analytic result (32) is expected to hold only in the hydrodynamic limit $t, \ell \to \infty$ with their ratio fixed. Indeed, upon increasing $\ell$ the data approach (32). Importantly, the fact that the initial state is a Fermi sea gives rise to logarithmic corrections. This will also happen for the entanglement entropies, as we will discuss in section 5.2. These corrections are visible for $M_2$ (see the inset in Fig. 7 (b)). In the figure we plot the deviation $\delta M_2$ from the hydrodynamic result, which is defined as

$$\delta M_2 := M_2^{\text{hydro}} - M_2. \tag{39}$$

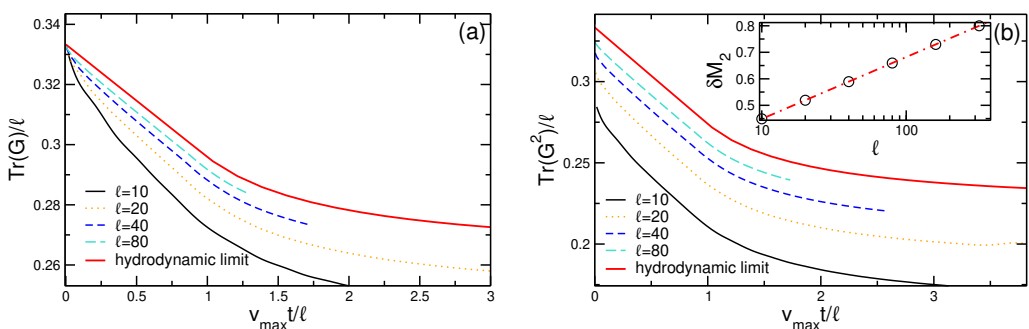

Figure 7: Moments of the fermionic correlator $M_n = \mathrm{Tr}(G^n)$ restricted to subsystem $A$ (bipartition in Fig 1 (a)). We show the rescaled moments $M_n/\ell$, with $\ell$ the size of $A$ plotted versus $v_{\max}t/\ell$. Here $v_{\max}$ is the maximum velocity. All the results are for $\gamma^- = 1$. The two panels are for $n = 1$ and $n = 2$. Different lines denote different subsystem size $\ell$. The dashed-dotted line is the analytic result in the hydrodynamic limit. Sizeable finite-time and finite-size corrections are present. In (b) we show the deviation from the hydrodynamic result at $t = 0$, $\delta M_2 = M_2^{\mathrm{hydro}} - M_2$ as a function of $\ell$. Notice the logarithmic scale on the $x$-axis.

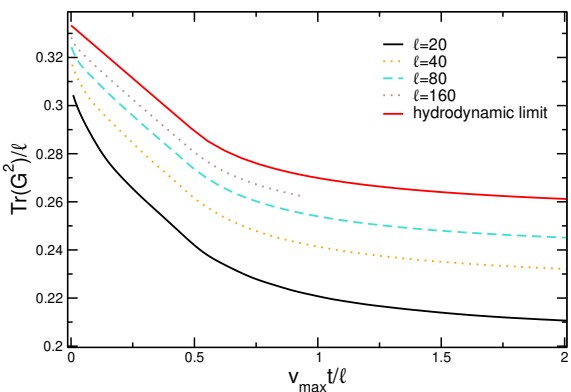

Figure 8: Same as in Fig. 7 for the interval $A'$ (centered partition in Fig 1 (b)).

We consider the initial deviation at $t = 0$. At $t = 0$ one expects that in the limit $\ell \to \infty$, $M_2 = k_f/\pi\ell$. The results in the inset of Fig. 7 (b) suggest the logarithmic behavior as

$$\delta M_2 = a_2 \ln(\ell) + \dots, \tag{40}$$

with the dots denoting subleading terms, and $a_2$ a constant. The dashed-dotted line in the inset of Fig. 7 (b) is obtained by fitting the constant $a_2$ in (40). The fit gives $a_2 \approx 0.101$. To our knowledge there is no analytic determination of the constant $a_2$, although it should be possible by using standard techniques for free-fermions systems. Moreover, although the data in Fig. 7 (b) suggest that such logarithmic terms survive at finite time, it is not clear a priori whether the constant $a_2$ remains the same. Finally, we should remark that the same logarithmic terms should be present for the centered partition in Fig. 1 (b). Indeed, for $t = 0$ the system is translational invariant, and the moments $M_n$ do not depend on the position of the subsystem. We discuss numerical results for $M_2$ for the centered partition (Fig. 1 (b)) in Fig. 8. As it is clear from the figure, the qualitative behavior is the same as for the side bipartition (see Fig. 7 (b)). Similar finite-size effects as in Fig. 7 (b) are present. Upon approaching the hydrodynamic limit $t, \ell \to \infty$ deviations from the

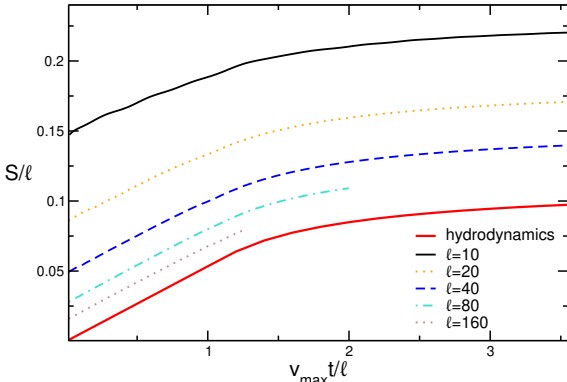

Figure 9: Entanglement entropy $S$ in the fermionic chain subjected to localized losses. We consider subsystem $A$ (bipartition in Fig. 1 (a)). The figure shows the entropy density $S/\ell$ plotted versus $v_{\max}t/\ell$, with $\ell$ the size of $A$ and $v_{\max}$ the maximum velocity. All the results are for fixed loss rate $\gamma^- = 1$ and $k_F = \pi/3$. We show results for several values of $\ell$, and the analytic result in the hydrodynamic limit (red continuous line in the figure).

hydrodynamic limit result (red continuous line in the figure) vanish.

## 5.2   Entanglement entropies and logarithmic negativity

Let us now discuss the out-of-equilibrium dynamics of the entanglement entropies. We first focus on the entanglement entropy for subsystem $A$ next to the dissipation source (as in Fig. 1 (a)). Our data are reported in Fig. 9. We restrict ourselves to fixed $\gamma^- = 1$, plotting the entropy density $S/\ell$ versus the rescaled time $v_{\max}t/\ell$. We show data for $\ell \in [10, 160]$. We also report the analytic result in the hydrodynamic limit (cf. (35)). Clearly, the numerical data exhibit the expected linear growth for $t \le \ell/(v_{\max})$, followed by a saturation at infinite time. Still, one should observe the sizeable deviations from the analytic result in the hydrodynamic limit (35). This is expected due to the finite $\ell$ and finite time $t$. Upon approaching the hydrodynamic limit, however, the deviation from (35) decrease. An important remark is that since the initial Fermi sea is a critical state, one should expect nontrivial finite-size corrections to the linear entanglement entropy growth. For instance, at $t = 0$ the entanglement entropies grow logarithmically with $\ell$ as in (36). In Fig. 10 we subtract the CFT contribution by plotting $S - 1/3 \ln(\ell)$. The data are the same as in Fig. 9. As it is clear from the figure, now the subtracted data exhibit a better agreement with the hydrodynamic result.

We perform a similar analysis for the Rényi entropies. In Fig. 11 we show numerical data for the second Rényi entropy $S^{(2)}$ plotted versus $v_{\max}t/\ell$. We only consider the bipartition in Fig. 1 (a). The data are for $\gamma^- = 1$ and the initial Fermi sea with $k_F = \pi/3$. As for the von Neumann entropy, we subtract the CFT contribution (cf. (36) with $n = 2$) that is present at $t = 0$. In the Figure we only show data for $v_{\max}t/\ell \lesssim 1$. The agreement with the analytic result in the hydrodynamic limit (35) is satisfactory.

Finally, it is crucial to compare the dynamics of the entanglement entropy with that of the logarithmic negativity (see section 3). As it was stressed in section 3 the entanglement entropies are not proper entanglement measures in the presence of dissipation because the full system is in a mixed state. On the other hand, the fermionic negativity $\mathcal{E}$ (cf. (28)) should be sensitive to genuine quantum correlation only.

As it was anticipated in the introduction, genuine entanglement and statistical correlations are deeply intertwined, but it is possible to distinguish them by comparing the

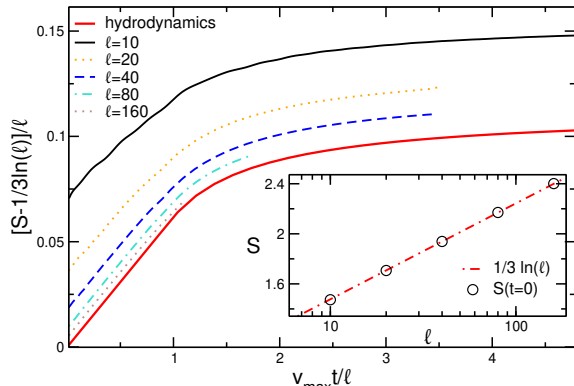

Figure 10: Same data as in Fig. 9 plotting $(S - 1/3\ln(\ell))/\ell$, where $1/3\ln(\ell)$ is the initial entanglement entropy. On the $x$-axis $v_{\max}$ is the maximum velocity and $\ell$ is the size of $A$. Inset: The entanglement entropy at $t = 0$ plotted versus $\ell$. Note the logarithmic scale on the $x$-axis. The dashed-dotted line is fit to the CFT prediction $1/3\ln(\ell) + a$, with $a$ a fitting constant.

behavior of the von Neumann entropy and of the logarithmic negativity for the two bi-partitions in Fig. 1 (a) and (b). Specifically, subsystem $A$ (see Fig. 1 (a)) is entangled with its complement because the reflected and the transmitted fermions, which form en-tangled pairs, are shared between them. Oppositely, this is not the case for $A'$ because the transmitted and the reflected fermions are never shared. This scenario implies that the entanglement entropy of $A$ and $A'$ exhibit a linear growth with time. On the other hand, only the logarithmic negativity of $A$ is expected to grow with time.

This is demonstrated in Fig. 12 (a) and (b). In Fig. 12 (a) we plot the rescaled negativity $\mathcal{E}/\ell$ versus the rescaled time $v_{\max}t/\ell$, whereas in Fig. 12 (b) we show the rescaled entanglement entropy. The data are for fixed $\gamma^- = 1$ and $k_F = \pi/3$ and subsystem' size $\ell = 160$. In both panels we show results for the subsystems $A$ (see Fig. 1 (a)) and $A'$ (see Fig. 1 (b)). It is clear from the figure that both the negativity and the von Neumann entropy of $A$ increase linearly with time. For the von Neumann entropy we report the expected slope of the linear growth in the hydrodynamic limit (dashed-dotted line in Fig. 12 (b)), which is in perfect agreement with the finite-size numerical results. Notice that at asymptotically long times the von Neumann entropy saturates (not shown in the figure), as already discussed in the previous sections. This saturation happens for the logarithmic negativity as well, as expected from the quasiparticle picture discussed above. This is shown explicitly in the inset in Fig. 12 (a) for subsystem $A$ of length $\ell = 20$. As in the main plot we show $\mathcal{E}/\ell$ versus $v_{\max}t/\ell$.

Let us now discuss the entanglement growth for the bipartition in Fig. 1 (b). The negativity (see Fig. 12), does not grow with time but it remains almost constant, showing a small decreasing trend at long times. Oppositely, the entanglement entropy exhibits a linear growth (see Fig. 12 (b)), which, again, does not reflect entanglement production. The slope of the linear growth (dashed-dotted line) is in agreement with (35).

## 6    Conclusions

We investigated the interplay between entanglement and statistical correlation in a uniform Fermi sea subjected to localized losses. We focused on the hydrodynamic limit of long

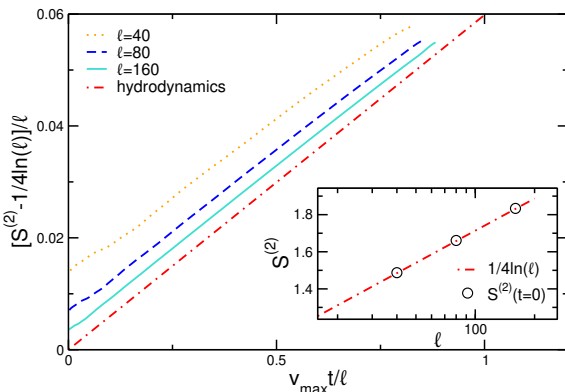

Figure 11: Out-of-equilibrium dynamics of the Rényi entropy $S^{(2)}$ of subsystem $A$ (bipartition in Fig. 1) (a). We plot the subtracted entropy $(S - 1/4\ln(\ell))/\ell$, with $1/4\ln(\ell)$ the Fermi sea entropy at $t = 0$. Data are for $\gamma^- = 1$ and $k_F = \pi/3$. On the $x$-axis $v_{max}$ is the maximum velocity and $\ell$ is the size of $A$. In the inset we show the entropy at $t = 0$ plotted versus $\ell$ to highlight the logarithmic increase.

times and the large subsystems, with their ratio fixed. In this regime the dynamics of the entanglement entropies can be understood analytically. We showed that the logarithmic negativity correctly diagnose the production of genuine quantum entanglement, whereas the entanglement entropies are sensitive to both quantum as well as classical correlation.

Let us now illustrate some interesting directions for future research. First, our results hold for the Fermi sea as initial state. It should be possible to generalize them to other situations, such as finite-temperature states, or inhomogeneous initial states, for instance, the domain-wall state. One should expect the linear entanglement growth to persist. An interesting possibility is to consider the out-of-equilibrium dynamics starting from product states. Thus, even in absence of losses the entanglement entropy grows linearly with time due to the propagation of entangled pairs of quasiparticles. It would be interesting to understand how this scenario is modified by localized losses. An interesting direction is to try to generalized the hydrodynamic framework to the logarithmic negativity, for which it should be possible to obtain a formula similar to (35).

Interestingly, our results suggest that local dissipation generically induces robust entanglement production. An important direction is to try to check this scenario for other types of local dissipation. An interesting candidate is incoherent hopping [89]. Unlike loss dissipation, for incoherent hopping the Liouvillian describing the dynamics of the density matrix is not quadratic. It would be interesting to understand whether the hydrodynamic approach outlined here still applies, at least in the weak dissipation limit. Moreover, an interesting direction would be understand the interplay between entanglement, local dissipation, and criticality [90]. Finally, it is important to investigate possible experimental verification of our results. Measuring entanglement in experiments is challenging, although recent results with cold-atom systems are promising [91], at least for Rényi entropies. On the other hand, the logarithmic negativity, which is a proper entanglement measure in the presence of dissipation, is not easy to measure with the current experimental tools. Fortunately, it is possible to detect genuine entanglement by using the moments of the partially transposed reduced density matrix, which are accessible experimentally [92].

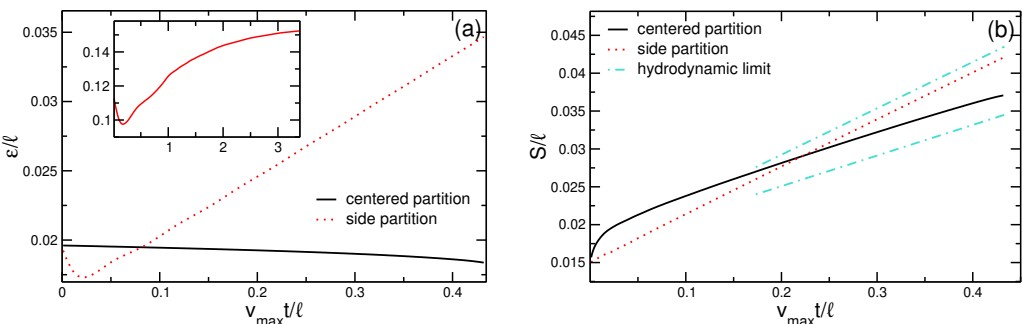

Figure 12: Comparison between the logarithmic negativity $\mathcal{E}$ and the entanglement entropy $S$ for the two bipartitions in Fig. 1 (a). Panels (a) and (b) show $\mathcal{E}/\ell$ and $S/\ell$ plotted versus $v_{\max}t/\ell$, respectively. Here $\ell$ is the size of $A$ and $A'$ (see Fig. 1) and $v_{\max}$ the maximum velocity. As it is clear from (a) the negativity $\mathcal{E}$ of $A$ exhibits a linear increase with time, whereas that of $A'$ depends mildly on time. Oppositely, the entanglement entropy of both $A$ and $A'$ increases linearly with time (see (b)).

## A  Entanglement entropies in the hydrodynamic limit: Derivation of Eq. (29)

In this section we derive formula (29). We employ a similar strategy as in Ref. [93]. Let us consider the interval $A = [0, \ell]$ (see Fig. 1 (a)). The main ingredient is the correlation matrix $G_{x,y}$ (cf. (11)) restricted to $A$, i.e., with $x, y \in A$. First, we can rewrite (11) as

$$G_{x,y} = \int_{-k_F}^{k_F} \frac{dk}{2\pi} S_{k,x} \bar{S}_{k,y}, \tag{41}$$

where we defined

$$S_{k,x} = e^{ikx} + r(k) e^{i|kx|} \int_{-\infty}^{\infty} \frac{dq}{2\pi i} \frac{e^{i(|v_k|t - |x|)q}}{q - i0}, \tag{42}$$

with $v_k$ the fermion group velocity (cf. (5)). The last term ensures the condition (14) and it relies on the well-known identity

$$\int_{-\infty}^{\infty} \frac{dq}{2\pi i} \frac{e^{iqx}}{q + i0} = \Theta(x), \tag{43}$$

where $i0$ is a positive convergence factor. Let us define $A_{k,q}$ as

$$A_{k,q}(t) := \frac{e^{it|v_k|q}}{q - i0} r(k). \tag{44}$$

To proceed we use the following identity

$$\sum_{z=1}^{\ell} e^{izk} = \frac{\ell}{4} \int_{-1}^{1} d\xi\, w(k) e^{i(\ell\xi + \ell + 1)k/2}, \quad \text{with } w(k) := \frac{k}{\sin(k/2)}. \tag{45}$$

Let us now define

$$F_{k_i, k_j} := F_{k_i, k_j}^{uu} + F_{k_i, k_j}^{ud} + F_{k_i, k_j}^{du} + F_{k_i, k_j}^{dd}, \tag{46}$$

with

$$F_{k_i,k_j}^{uu} := \frac{\ell}{4} \int d\xi w(k_i - k_j) e^{i\ell(\xi+1)(k_i-k_j)/2} \tag{47}$$

$$F_{k_i,k_j}^{ud} := -\frac{\ell}{4} \int d\xi \int \frac{dq}{2\pi i} w(k_i - |k_j| + q) e^{i\ell(k_i-|k_j|+q)(\xi+1)/2} \bar{A}_{k_j,q} \tag{48}$$

$$F_{k_i,k_j}^{du} := \frac{\ell}{4} \int d\xi \int \frac{dq}{2\pi i} w(|k_i| - k_j - q) e^{i\ell(|k_i|-k_j-q)(\xi+1)/2} A_{k_i,q}, \tag{49}$$

and

$$F_{k_i,k_j}^{dd} := -\frac{\ell}{4} \int d\xi \int \frac{dq'}{2\pi i} \int \frac{dq}{2\pi i} w(|k_i| - |k_j| - q + q')$$
$$\times e^{i\ell(|k_i|-|k_j|-q+q')(\xi+1)/2} A_{k_i,q} \bar{A}_{k_j,q}. \tag{50}$$

To derive (29), it is convenient to consider the moments of the correlation matrix $M_n = \text{Tr}(G^n)$. For generic integer $n$, by using (41), (46) and (47)-(50) one obtains that

$$\text{Tr}(G^n) = \int_{-k_F}^{k_F} \frac{d^n k}{(2\pi)^n} \prod_{i=1}^{n} F_{k_i,k_{i-1}}. \tag{51}$$

Here the variables $k_i$ are arranged in cyclic order, i.e., $k_0 = k_n$. We can rewrite $F_{k_i,k_j}^{\alpha,\beta}$, with $\alpha, \beta = u, d$ (cf. (47)-(50)) as

$$F_{k_i,k_j}^{uu} = \frac{\ell}{2} \int d\xi e^{i\ell(\xi+1)(k_i-k_j)/2} \tag{52}$$

$$F_{k_i,k_j}^{ud} = \frac{\ell}{2} \int d\xi e^{i\ell(k_i-|k_j|)(\xi+1)/2} r(k_j) \Theta(-\ell(\xi+1)/2 + |v_{k_j}|t) \tag{53}$$

$$F_{k_i,k_j}^{du} = \frac{\ell}{2} \int d\xi e^{i\ell(|k_i|-k_j)(\xi+1)/2} r(k_i) \Theta(-\ell(\xi+1)/2 + |v_{k_i}|t), \tag{54}$$

and

$$F_{k_i,k_j}^{dd} = \frac{\ell}{2} \int d\xi e^{i\ell(|k_i|-|k_j|)(\xi+1)/2} r(k_i) r(k_j)$$
$$\times \Theta(-\ell(\xi+1)/2 + |v_{k_i}|t) \Theta(-\ell(\xi+1)/2 + |v_{k_j}|t). \tag{55}$$

To obtain (52)-(55), we used that in the hydrodynamic limit $\ell, t \to \infty$ with the ratio $t/\ell$ fixed the integrals in (51) are dominated by the regions with $k_i \to k_j$ and $q \to 0$. This implies that $w(k_i - k_j) \to 1/2$ (cf. (45)), and that one can perform the integration over $q$ and $q'$ in (47)-(50), which, by using (43), give the Heaviside theta functions in (52)-(55). We can now rewrite (51) as

$$\text{Tr}(G^n) = \left(\frac{\ell}{2}\right)^n \int_{-k_F}^{k_F} \frac{d^n k}{(2\pi)^n} \int_{-1}^{1} d^n \xi \prod_{i=1}^{n} \widetilde{F}_{k_i,k_{i-1}}(\xi_i). \tag{56}$$

Here we defined $\widetilde{F}_{k_i,k_{i-1}} := \widetilde{F}_{k_i,k_j}^{uu} + \widetilde{F}_{k_i,k_j}^{ud} + \widetilde{F}_{k_i,k_j}^{du} + \widetilde{F}_{k_i,k_j}^{dd}$, where $\widetilde{F}_{k_i,k_j}^{\alpha,\beta}$ with $\alpha, \beta = u, d$ are the integrands appearing in (52)-(55). To proceed, we now treat the integrals over $\xi_i$ by using the stationary phase approximation in the hydrodynamic limit. We first observe that (56) can be rewritten as

$$\text{Tr}(G^n) = \left(\frac{\ell}{2}\right)^n \int_{-k_F}^{k_F} \frac{d^n k}{(2\pi)^n} \int_{-1}^{1} d^n \xi$$
$$\sum_{\sigma_i, \tau_i = 0,1} \prod_{i=1}^{n} e^{i\ell(\xi_i+1)(k_{\sigma_i}-k_{\tau_{i-1}})/2} \tilde{r}^{\sigma_i}(\xi_i, k_{\sigma_i}) \tilde{r}^{\tau_{i-1}}(\xi_i, k_{\tau_{i-1}}). \tag{57}$$

Here we defined

$$\tilde{r}(\xi, k) = r(k)\Theta(-\ell(\xi+1)/2 + |v_k|t). \tag{58}$$

Here we also defined

$$k_{\sigma_i} = \begin{cases} k_i & \text{if } \sigma_i = 0 \\ |k_i| & \text{if } \sigma_i = 1 \end{cases} \tag{59}$$

The same definition as (59) holds for $k_{\tau_i}$. Eq. (57) arises directly from (56). Each term $\widetilde{F}_{k_i,k_j}$ in (56) contains a phase factor $e^{i\ell(\xi_i+1)(k_i-k_{i-1})}$, where $k_i$, $k_{i-1}$ can be replaced by $|k_i|, |k_{i-1}|$. Each term $|k_i|$ is accompanied by a factor $\tilde{r}(\xi_i, k_i)$. The sum over $\sigma_i, \tau_i$ in (57) accounts for all the possible ways of distributing the terms with the absolute values $|k_i|$.

We first focus on the situation with $\sigma_i = \tau_i = 0$ for any $i$. Thus, within the stationary phase approximation the integral in (57) in the large $\ell$ limit is dominated by the stationary points of the exponent of the phase factor. By imposing stationarity with respect to the variables $\xi_i$, one obtains that

$$k_i = k_1, \quad \forall i. \tag{60}$$

Now (57) becomes

$$\mathcal{I}_0^{(n)} = \left(\frac{\ell}{2}\right)^n \int_{-k_F}^{k_F} \frac{d^n k}{(2\pi)^n} \int_{-1}^{1} d^n \xi e^{i\ell \sum_{i=1}^{n} (\xi_i+1)(k_i-k_{i-1})/2}. \tag{61}$$

Although the integral (61) can be computed exactly, it is useful to discuss the stationary phase approximation. Let us change variables as

$$\zeta_1 := \xi_1 \tag{62}$$
$$\zeta_i := \xi_{i+1} - \xi_i, \quad i \in [1, n]. \tag{63}$$

The variables $\xi_i$ and $\zeta_i$ satisfy cyclic boundary conditions. We obtain that (61) is rewritten as

$$\mathcal{I}_0^{(n)} = \left(\frac{\ell}{2}\right)^n \int_{-k_F}^{k_F} \frac{d^n k}{(2\pi)^n} \int d^n \zeta e^{-i\ell \sum_{j=1}^{n} \zeta_j(k_j-k_1)/2}. \tag{64}$$

Notice that the 1 in $(\xi_i + 1)$ in (61) cancels out in the sum over $i$, and it would also be irrelevant at the stationary point where $k_i \to k_1$ for any $i$. The integrand in (64) does not depend on $\zeta_1$. The integration domain for the variables $\zeta_1$ is given as

$$-1 \leq \zeta_1 - \sum_{j=k}^{n} \zeta_j \leq 1, \quad \forall k \in [2, n]. \tag{65}$$

As the integrand in (64) does not depend on $\zeta_1$, one can perform the integral to obtain

$$\mathcal{I}_0^{(n)} := \left(\frac{\ell}{2}\right) \int_{-k_F}^{k_F} \frac{dk_1}{2\pi} \Lambda_0^{(n-1)}(k_1) =$$
$$\left(\frac{\ell}{2}\right)^n \int_{-k_F}^{k_F} \frac{dk_1}{2\pi} \int_{-k_F}^{k_F} \frac{d^{n-1}k}{(2\pi)^{n-1}} \int d^{n-1}\zeta e^{-i\ell \sum_{j=1}^{n} \zeta_j(k_j-k_1)/2} \mu(\{\zeta_k\}), \tag{66}$$

where we also isolated the integration over $k_1$. Here $\mu(\{\zeta_k\})$ is the measure of the allowed values for $\zeta_1$, and it reads as

$$\mu(\{\zeta_k\}) = \max\left[0, \min_{k\in[2,n]}\left[1 - \sum_{j=k}^{n} \zeta_j\right] + \min_{k\in[2,n]}\left[1 + \sum_{j=k}^{n} \zeta_j\right]\right]. \tag{67}$$

We can now apply the stationary phase approximation to the integral

$$\Lambda_0^{(n-1)} = \left(\frac{\ell}{2}\right)^{n-1} \int_{-k_F}^{k_F} \frac{d^{n-1}k}{(2\pi)^{n-1}} \int d^{n-1}\zeta \, e^{-i\ell \sum_{j=1}^{n} \zeta_j(k_j - k_1)/2} \mu(\{\zeta_k\}). \tag{68}$$

The stationary phase approximation states that for sufficiently smooth $N$-dimensional functions $f(\boldsymbol{x})$ and $g(\boldsymbol{x})$, in the limit $\ell \to \infty$ one has [88]

$$\int_\Omega d^N x \, g(\boldsymbol{x}) e^{i\ell f(\boldsymbol{x})} = \left(\frac{2\pi}{\ell}\right)^{N/2} g(\boldsymbol{x}_0) |\det H|^{-1/2} e^{i\ell f(\boldsymbol{x}_0) + \frac{i\pi\sigma}{4}}, \tag{69}$$

where $\Omega$ is the integration domain, $\boldsymbol{x}_0$ is the stationary point of $f(\boldsymbol{x})$, i.e., such that $\nabla f(\boldsymbol{x}) = 0$, $H$ is the Hessian matrix, and $\sigma$ its signature, i.e., the difference between the number of positive and negative eigenvalues. A straightforward application of the stationary phase gives that in the limit $\ell \to \infty$ the $\Lambda_0^{(n-1)}$ is dominated by the stationary point as

$$\bar{k}_j = k_1, \quad j = 2, \ldots, n, \tag{70}$$
$$\bar{\zeta}_j = 0, \quad j = 2, \ldots, n. \tag{71}$$

In our case the phase in (69) vanishes and the signature of the Hessian is zero. Moreover, $\det H = 2^{-2n+2}$. Putting everything together we obtain that

$$\Lambda_0^{(n-1)} = 2, \tag{72}$$

where we used that $\mu(\{\zeta_k\}) = 2$ at the stationary point. Note that there is no dependence on $k_1$ in (72). Finally, we obtain that the integral (64) is given as

$$\mathcal{I}_0^{(n)} = \ell \int_{-k_F}^{k_F} \frac{dk_1}{2\pi}. \tag{73}$$

Let us now consider the generic integral (57). We now observe that for any pair of indices $(\sigma_i, \tau_i)$ there are two possible situations that can occur. Specifically, we define $(\sigma_i, \tau_i)$ as *paired* if $\sigma_i = \tau_i = 1$, whereas we define $(\sigma_i, \tau_i)$ as *unpaired* otherwise. Notice that if $(\sigma_i, \tau_i)$ are paired it means that both occurrences of momentum $k_i$ appear with the absolute value $|k_i|$ in (57).

It is straightforward to convince oneself that the presence of a single set of unpaired indices $(\sigma_i, \tau_i)$ implies that in the limit $\ell \to \infty$ the stationary point is given as

$$k_i = k_1 > 0, \quad \forall i, \tag{74}$$

i.e., all the momenta have to be positive to have a finite contribution in the stationary phase. This implies that one can remove the absolute values of the momenta. Then, the derivation of the stationary phase is similar to that for the case with $\sigma_i = \tau_i = 0, \forall i$.

An important difference is that for any index $\sigma_i = 1$ and $\tau_i = 1$ there is a factor $\tilde{r}(\xi_i, k_i)$. This implies that the integration over $\zeta_1$ in principle cannot be performed as in (72). However, at the stationary point, from (71) one obtains that $\xi_i \to \xi_1 = \zeta_1$ and $k_i \to k_1$ for any $i$. One is left with the integral over $\zeta_1$ as

$$\int_{-1}^{1} d\zeta_1 \Theta(-\ell(\zeta_1 + 1)/2 + t|v_k|) = 2\min(|v_k|t/\ell, 1), \quad \text{with } k > 0. \tag{75}$$

Let us now discuss what happens when paired indices are present. It is clear that the main consequence of the presence of paired indices $\sigma_i, \tau_i$ is a factor 2 because the integrands do

not depend on the sign of the momenta. To discuss the result of the stationary phase, let us define for the following the number of paired momenta as $p$, and the total number of momenta appearing with the absolute value as $n_a$. Let us consider the case $n_a > 0$ since the case with $n_a = 0$ was treated above. $n_a$ is the number of $\sigma_i = 1$ and $\tau_i = 1$. The number of terms $N_{(n,n_a,p)}$ with fixed $n, n_a, p$ can be obtained by elementary combinatorics as

$$N_{(n,n_a,p)} = \binom{n}{p}\binom{n-p}{n_a - 2p} 2^{n_a - 2p}. \tag{76}$$

Now we use take into account that for each set of paired indices there is a factor two. By summing over the possible number of pairs $p$, we obtain the total number of terms $N'_{(n,n_a)}$ as

$$N'_{(n,n_a)} = \sum_{p=0}^{\lfloor n_a/2 \rfloor} 2^p N_{(n,n_a,p)}. \tag{77}$$

We now use that for each $n_a$ there is term $r^{n_a}$. Finally, it is straightforward to perform the sum over $n_a$ to obtain the total contribution as

$$\mathrm{Tr}(G^n) = \left(\frac{\ell}{2}\right) \int_{-k_F}^{k_F} \frac{dk_1}{2\pi} (\Lambda_0^{(n-1)} + \Lambda^{(n-1)}) \tag{78}$$

where $\Lambda_0^{(n-1)} = 2$ and

$$\Lambda^{(n-1)} = 2\min(|v_{k_1}|t/\ell, 1)\left(\sum_{n_a=1}^{2n} N'_{(n,n_a)} r^{n_a} - 1\right)\Theta(k_1). \tag{79}$$

We now use that

$$\sum_{n_a=1}^{2n} N'_{(n,n_a)} r^{n_a} = (1 + 2r + 2r^2)^n - 1. \tag{80}$$

where $r(k_1)$ is the reflection amplitude in (12). We can also use that

$$1 + 2r + 2r^2 = 1 - |a|^2, \tag{81}$$

where $|a(k)|^2$ is the absorption coefficient. Thus, putting everything together one obtains the final formula for $\mathrm{Tr}(G_A^n)$ as

$$\mathrm{Tr}(G_A^n) = \ell \int_{-k_F}^{k_F} \frac{dk}{2\pi}\left[1 - \frac{1}{2}\min(|v_k|t/\ell, 1) + \frac{1}{2}(1 - |a(k)|^2)^n \min(|v_k|t/\ell, 1)\right]. \tag{82}$$

Here we replaced $k_1 \to k$ and we used the fact that the integrand is symmetric under $k \to -k$ to remove the factor $\Theta(k)$ in (79). The subscript $A$ in (82) is to stress that it holds for subsystem $A$ (see Fig. 1 (a)). Crucially, Eq. (82) depends only on the local density of fermions $1 - |a|^2$ that are not absorbed at the origin.

Finally, we comment on the modifications in order to generalize (82) to the case of the bipartition in Fig. 1 (b), i.e., for the interval $A'$ centered around the impurity. The main difference is that (77) has to be replaced by $\widetilde{N}'_{(n,n_a)}$ as

$$\widetilde{N}'_{(n,n_a)} = \frac{2^{n_a}}{4}\binom{2n}{n_a}. \tag{83}$$

A straightforward generalization of the steps leading to (82) gives

$$\mathrm{Tr}(G_{A'}^n) = \ell \int_{-k_F}^{k_F} \frac{dk}{2\pi}\left[1 - \frac{1}{4}\min(2|v_k|t/\ell, 1) + \frac{1}{4}(1 - 2|a(k)|^2)^n \min(2|v_k|t/\ell, 1)\right]. \tag{84}$$

The factor $1/4$ and the $2|v_k|$ in the integrand in (84) reflect that the distance from the impurity and the edges of subsystem $A'$ is $\ell/2$ and not $\ell$ as in (82). Furthermore, the factor $1 - 2|a|^2$ instead of $1 - |a|^2$ in (82) arises because fermions are absorbed from both sides of the impurity.

Finally, by useing (82) and (84) one can obtain the hydrodynamic behavior of

$$\text{Tr}(\mathcal{F}(G)) \tag{85}$$

where $\mathcal{F}(z)$ is smooth enough to admit a Taylor expansion around $z = 0$. By Taylor expanding $\mathcal{F}(z)$ for $z = 0$ and by using (82) and (84), one obtains (29).

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
