# Peer review of "Unbounded entanglement production via a dissipative impurity"

_SciPost Physics_

## Round 1 · Referee Report · Anonymous · 2021-10-22

Report

I would like to thank the author for responding in detail to my questions and for taking my suggestions into account. I recommend this manuscript for publication without further modifications.

---

## Round 1 · Author Response

Dear Editor,

I would like to thank you for your work and the referees for the positive evaluation of my paper and for their comments and suggestions. Here below I address all the points that they raised in their reports.

Sincerely Yours, Vincenzo Alba

**REPLY TO REFEREE II

REFEREE: The paper presents results on entanglement dynamics of a one dimensional chain of fermions with a single dissipative impurity. The problem is thoroughly studied from both numerical and analytical perspectives. The paper is well-organized, clearly written and extensively referenced. The results are of modest interest as the problem exhibits different regimes of entanglement dynamics with a simple physical picture. As a result, I recommend the paper for publication. To strengthen the results, I recommend the author adds additional discussion of the experimental realizations and probes of this phenomena in cold atom or solid-state systems.

REPLY: I thank the referee for the positive evaluation of my work. As for the experimental realizations, there are several experiments with cold atoms that are close to the setup investigated in the paper. For instance, I now mention Ref.[53,54] where the authors study the effect of localized losses on quantum transport in one dimensional fermionic systems. As it clear the setup is very similar to the one that I investigate in the paper. Moreover, although in Ref.[53,54] the fermions are interacting, the interactions are weak and in any case they in principle can be tuned. As for the possibility to measure entanglement, this is in general challenging. A possible strategy is to use the results of Ref [91], which allows to measure experimentally the R\'enyi entropies. Notice that, however, the logarithmic negativity is still out of reach. This makes challenging to distinguish genuine quantum entanglement from the spurious correlations that arise from the fact that the system is in a mixed state. Fortunately, alternative quantities exist which are based on the moments of the partially transposed reduced density matrix, and that can be measured experimentally, as described in Ref. [92] that I now mention in the Conclusions.

**REPLY TO REFEREE I

REFEREE: In this work, the author considers entanglement production by a single dissipative impurity in one-dimensional noninteracting systems. The dissipation is modeled by the Lindblad equation, and information about entanglement and the logarithmic negativity is extracted from the covariance matrix, i.e, two-point fermionic correlators. In the hydrodynamic limit, i.e., at long times and large distances, and starting from the filled Fermi Sea, the author finds that the entanglement entropy increases linearly at short times before saturating to a value that satisfies the volume law. Comparing this behavior to the logarithmic negativity, the author concludes that a certain extent of the entanglement production does not originate from "genuine" entanglement production, but from classical correlations between the two considered subsystems, a result of the system being in a mixed state due to dissipation.

The manuscript is scientifically sound, well-written, and well-structured. The results are presented in a clear and comprehensive 
way and shine light onto the role of entanglement in dissipative systems. I have only a few remarks the author should consider prior to publication:

REPLY: I thank the referee for the positive assessment and for the useful comments.

REFEREE: As motivated in the introduction, the author chose the single-impurity setup as a minimal model to study the effects of dissipation in many-body systems. However, the considered system is noninteracting. I would welcome a comment if the author has an intuition about the effect of interactions.

REPLY: I thank the referee for this question. This is a very challenging problem. A plausible scenario is that if the model is integrable and interacting, away from the dissipation source the dynamics is described by the quasiparticles of the integrable bulk model. The problem is reduced then to determining the scattering properties of the impurity. It is not clear a priori that this problem can be solved analytically, even in the situation in which the total system, including the impurity is integrable. Physically, however, it is natural to expect genuine entanglement production also in the interacting case, the reason being the presence of a reflected and transmitted wave around the impurity.

REFEREE: 2. Are there any experimental observables linked to the findings of this paper? While the von-Neumann entropy is notoriously difficult to measure experimentally, I am wondering if there are indirect measures sensitive to the production of entanglement, especially observables that distinguish between "genuine" entanglement and classical entanglement.

REPLY: I thank the referee for this question. As I now mention in the Conclusions, while the von Neumann entropy and the negativity are very hard to measure in experiments, very recently it has been shown that the moments of the partially transposed reduced density matrix can be used to detect genuine entanglement. These moments can be measured experimentally with cold atoms, as done in Ref. [92].

REFEREE: 3. Related to the former point: The author might want to comment on the distinction between the genuine quantum and classical entanglement.

REPLY: This is a very important point. For mixed states the von Neumann, the Renyi entropies and the mutual information are not proper entanglement measures. To understand that, one can think of a mixed state as emerging from an enlarged pure state
(original system plus environment). By taking the trace over the environment one obtains the mixed density matrix of the original system. However, the trace operation introduces some classical correlation between A and its complement. The same problem occurs if the global system is in a pure state but one is interested in the entanglement between two non-complementary subsystems. I realized that this was not stressed enough in the manuscript. I now added this discussion in the Introduction.

REFEREE: 4. Some of the results overlap with the author's previous work, Ref. [44]. For example, Fig. 2 is already contained in Ref. [44] and the expressions for the covariance matrix are the same. The paper would benefit from disentangling the new contributions from previous work.

REPLY: I thank the referee for this observation. The overlap with Ref [44] is minimal. In the paper I use from Ref. [44] only formula (11), i.e., the expression of the correlation matrix in the hydrodynamic limit. I now changed the title of section 2 to ``Localized losses in a Fermi sea: Review of known results'' to stress that the results of section 2 have been derived elsewhere.

---

## Round 1 · List of Changes

1) Added References 53,54,91,92
2) Modified introduction to discuss the difference between entanglement and classical correlation due to the mixedness of the global state.
3) Changed title of section 2.
4) Modified Conclusions to discuss experimental implications of my work.

You are currently on this page

Resubmission scipost_202110_00009v1 on 8 October 2021

---

## Editorial Decision

publication_decision_taken:_accept